# Fatal Case of Progressive Mpox in a Patient with AIDS—Viral Enteropathy and Malabsorption Demanding the Use of Full Parenteral ARV and Endovenous Cidofovir

João Caria [1,*], Francisco Vara-Luiz [2,3], Inês Maia [4], Anneke Joosten [4], Luís Val-Flores [5], Hélder Pinheiro [1,6], Diana Póvoas [1,7], Nuno Germano [5] and Fernando Maltez [1,6,8,9]

1. Department of Infectious Diseases, Hospital de Curry Cabral, Centro Hospitalar Universitário de Lisboa Central, 1069-166 Lisboa, Portugal; helderp72@gmail.com (H.P.); d.povoas@gmail.com (D.P.)
2. Department of Gastroenterology, Hospital Garcia de Orta, 2805-267 Almada, Portugal; franciscovaraluiz@gmail.com
3. Grupo de Patologia Médica, Nutrição e Exercício Clínico (PaMNEC), Centro de Investigação Interdisciplinar Egas Moniz (CiiEM), 2829-511 Almada, Portugal
4. Department of Internal Medicine, Centro Hospitalar Barreiro-Montijo, 2830-003 Barreiro, Portugal; inesferreiramaia@gmail.com (I.M.); anneke.joosten@gmail.com (A.J.)
5. Department of Intensive Care Medicine, Hospital de Curry Cabral, Centro Hospitalar Universitário de Lisboa Central, 1069-166 Lisboa, Portugal; nuno.germano@chlc.min-saude.pt (N.G.)
6. Faculdade de Ciências Médica, Universidade Nova de Lisboa, 1169-056 Lisboa, Portugal
7. Instituto Gulbenkian de Ciência, 2780-156 Oeiras, Portugal
8. Instituto de Saúde Ambiental (ISAMB), Faculdade de Medicina, Universidade de Lisboa, 1600-190 Lisboa, Portugal
9. Laboratório Associado TERRA, Faculdade de Medicina, Universidade de Lisboa, 1600-190 Lisboa, Portugal
* Correspondence: caria@campus.ul.pt

**Abstract:** We report a fatal case of disseminated mpox infection that progressed over more than three months in an HIV-infected patient with acquired immunodeficiency syndrome (AIDS). Mucocutaneous, pleuropulmonary, central nervous system, and gastrointestinal involvement was documented. This course of disease resembles progressive vaccinia, a formerly reported disease caused by uncontrolled replication of smallpox vaccination orthopoxviruses in immunosuppressed patients. Severe small bowel involvement jeopardized normal oral tecovirimat and antiretroviral therapy absorption. This problem prompted the use of full parenteral antiretrovirals and endovenous cidofovir. Although a remarkable decrease in HIV viral load occurred in six days, mpox infection continued to progress, and the patient died of septic shock. This case offers new clinical insights on the presentation of severe disease in AIDS patients. Moreover, this case alerts for the need for prompt therapy initiation in patients at risk of ominous clinical progression.

**Keywords:** mpox; AIDS; HIV; progressive; enteropathy; fatal; cidofovir; tecovirimat

## 1. Introduction

Since the start of the current mpox (formerly monkeypox) outbreak, and as of 6 January 2023, 84,330 confirmed cases have been reported from 110 countries. Most of the patients are young males, and sexual contact is thought to be the main route of transmission. The majority of cases have a benign course. Hospital admission occurs in 7.3%, intensive care unit (ICU) admission in only 0.3% of the cases, and there have been 74 reported deaths [1].

Severe presentation with hospitalization and/or death has been reported more frequently in immunosuppressed individuals [2]. Possible exposure of HIV-infected individuals to mpox, especially those not on antiretroviral therapy (ART) and with low TCD4+ lymphocyte counts and high viremia, has raised concerns on ominous outcomes in this group. Indeed, cases of exuberant severity in immunosuppressed patients have been reported in the current outbreak, with remarkable cases of progressive gangrene despite the initiation

of ART and drugs with activity against mpox [3]. Complications also include sepsis due to secondary bacterial infection, gastrointestinal involvement, bronchopneumonia, corneal infection resulting in loss of vision, and encephalitis [4,5]. Therapeutic options remain scarce, with both tecovirimat and cidofovir presenting very limited efficacy data in humans [6,7].

Although antiretrovirals generally demonstrate excellent absorption, there are well-documented pharmacokinetic reports on ineffective plasma drug concentrations in patients with established infectious enteropathies [8].

We report a fatal case of disseminated and progressive mpox infection and a concurrent HIV infection with acquired immunodeficiency syndrome (AIDS) criteria, in which severe small bowel involvement jeopardized normal therapy absorption for both conditions.

## 2. Case Report

We report the case of a 23-year-old Brazilian man with a medical history of HIV-1 infection diagnosed in 2019 in the context of hospitalization for pulmonary pneumocystosis and cytomegalovirus (CMV) esophagitis and encephalitis. At that time, the patient had a lymphocyte TC4+ count of 56/μL. Opportunistic infections were treated accordingly, and ART was started with darunavir/cobicistat/tenofovir alafenamide/emtricitabine. No sequelae resulted from that clinical episode. After discharge, the patient had a period of fair ART adhesion with undetectable HIV viral load and a lymphocyte TC4+ count zenith of 471/μL in 2019. At this time, ART regimen was switched to dolutegravir/abacavir/lamivudine. After this period, the patient had irregular ART adhesion, reporting a 12-month total non-compliance to ART that lasted until one month prior to the first medical observation of this episode. His history was also remarkable for active recreational consumption of non-injectable methamphetamines and cocaine.

The patient sought medical attention to the emergency department of a district hospital in Barreiro, Portugal (day 0 upon beginning of the episode), in 2022 for a one-month duration clinical syndrome of bloody diarrhea, nausea, vomiting, and significant weight loss (>10%; the patient lost 10 kg from an initial weight of 65 kg). Additionally, he had a papular-pustular-vesicular rash with one-week time evolution with preferential face, mouth, axillar, scalp, genital, perineal, and anal location. The patient reported unprotected sex contacts with female sex workers in the weeks prior to the beginning of the rash.

On examination, the patient had fever with an axillary temperature of 38.0 °C; most of the cutaneous lesions had an umbilicated necrotic center or were ulcerated with a purulent exudate. The most severe ulcers in size (>2 cm diameter) and depth were located in the right genian region, penile gland and urinary meatus, scrotum, anal, and perianal region, with erosion of mucosa and submucosa and erythematous regular border. He complained of severe pain caused by anal and perianal ulcers. Lesion exudate samples were collected for mpox virus polymerase chain reaction (PCR), empirical oral antibiotics were started for probable bacterial coinfection, and the patient was discharged home. Upon reevaluation on day +9 as an outpatient, and by then with confirmation of positive PCR result for mpox, the patient was hospitalized in an internal medicine ward due to complicated mpox with insufficient pain management; progressive worsening of muco-cutaneous mpox lesions in number, size, depth, and exudation with presumptive bacterial superinfection (Figure 1A–C); and persistence of severe gastrointestinal symptoms without clear etiology.

HIV infection staging upon admission showed a TCD4+ lymphocyte count of 173/μL and an HIV-1 viral load of 36,904 copies/mL. Resistance testing (with results being available only on day +45) only documented M184V mutation in the reverse transcriptase region. He had no other remarkable lab results than mild anemia (hemoglobin of 11.2 g/dL) and C-reactive protein of 8.6 mg/L.

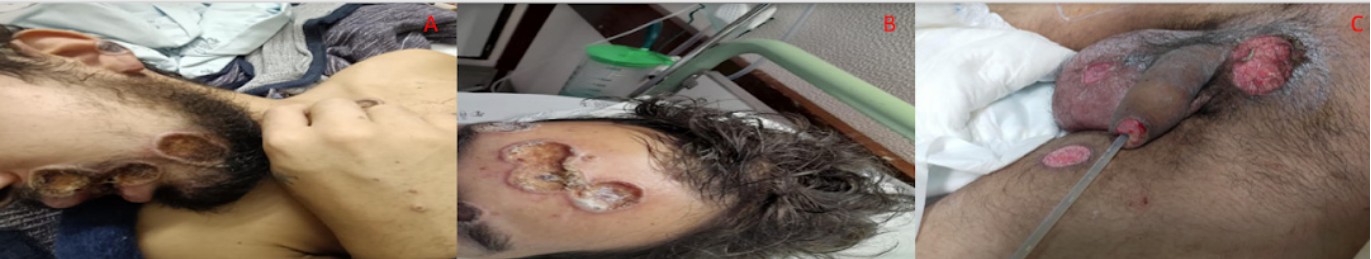

**Figure 1.** Clinical and imaging pictures of the case. (**A**–**C**) Pictures of perioral, frontal, and genital lesions at the beginning of hospitalization (day +9).

Empirical endovenous ganciclovir was initiated at admission, admitting simultaneous interplay of CMV colitis considering the presentation with diarrhea with bloody stool and previous history of CMV gastrointestinal disease. The high clinical suspicion and the late access to the result of negative CMV viral load justified that this therapy was carried out until day +30. Despite this antiviral empirical therapy and maintaining broad-spectrum parenteral antibiotic coverage (from day 0 to day +41), the patient maintained bloody diarrhea. Abdominopelvic computed tomography (CT) was performed on days +12 and +24, showing parietal thickening of the distal rectum and anus and marked diffuse densification in all the adipose planes of the perineum and buttocks as well as an area suggestive of a periluminal abscess in that region.

Anuscopy was performed on day +17 with the following findings: "Deep perianal coalescing ulcerations were observed, with a devitalized bottom and hardened border (Monkeypox lesions?), with adjacent darkened skin areas suggestive of islets of spared skin. These ulcerations extend circumferentially through the anal canal and inferior rectum in an extension of about 5 cm in depth, with greater exuberance and infiltration at 7–8 am, without evidence of rectal bulges on palpation". Drainage was attempted but failed due to lack accessible and drainable collection. Colonoscopy was attempted on day +25 but interrupted for lack of correct preparation. Further colonoscopic exams were avoided due to the risk of perforation. Anal border lesions were biopsied on day +30 with anatomopathological diagnosis of "exanthematic anal lesion compatible with monkeypox virus infection in the pustular phase" and microscopic description of "tissue with focus of necrosis of the entire thickness of the epithelial layer, (…) spongiosis of the epithelial layer, ballooning degeneration of these cells, (…) spherical intracytoplasmic inclusions of Guarnieri, along with "empty" nuclei. Direct exam, NAAT and mycobacterial culture of this tissue were negative".

Further etiological study of colitis by other microorganisms did not yield positive results. Bacterial coprocultures (with valorization of *Campylobacter* spp., *Salmonella* spp., *Shigella* spp., and Clostridium difficile growth); feces bacterial and fungi antigen testing (including *Giardia* spp., *Clostridioides* spp., *Isospora* spp., *Mycrosporidium* spp., and *Mycrosporidium*); feces viral antigen testing (Norovirus, Astrovirus, Norwalk); feces parasitological exam (search for eggs, cysts, and parasites); *Strongyloides* spp., *Entamoeba* spp., and VDRL serologies; CMV serum PCR; rectal exudate PCR for Chlamydia trachomatis and Neisseria gonorrhoeae were performed, and all these exams were negative. Autoimmune markers related to inflammatory bowel disease were also negative.

During this first month of hospitalization, and despite the therapies mentioned, in addition to gastrointestinal worsening, there was sustained clinical deterioration, with persistent fever and development of new and progression of all previous mucocutaneous lesions in size, depth, pain, and purulence, particularly in the anorectal region. From day +23, the patient developed dyspnea and hypoxemia with need of supplemental oxygen. Thoracic CT documented parenchymal nodularities in the right middle lobe and posterior basal segments of both lower lungs, with a maximum mass size of 11 mm. Bronchofibroscopy was performed on day +24 with description of diffuse hyperemia in bronchial mucosa and mucopurulent secretions bilaterally. Bronchoalveolar lavage was recovered—direct

and cultural bacterial, mycological, and mycobacterial exams were negative as well as Pneumocystis jirovecii PCR.

At the end of the first month of hospitalization, in addition to the above-mentioned clinical progression of all mucocutaneous lesions, mpox PCR on lesion exudate on day +31 kept a positive result. Pain management due to anal ulcers remained insufficient despite local and parenteral non-opiate and opiate analgesia as well as pain modulation adjuvant therapy. The severity of this symptom mentally impaired the individual to the extent of him attempting suicide on two occasions. Lumbar puncture was performed on day +34, and CSF was unremarkable.

As fever, hypoxemia and raising of inflammatory parameters persisted without any additional agent identification, and after 41 days of empirical antibiotics, thoracic CT was repeated on day +43 (Figure 2). The exam demonstrated multiple homogeneous nodular lesions scattered throughout the parenchyma with increase in number and lesion volume. Even though broad-spectrum antibiotherapy was reintroduced that same day and ARV initiated with bictegravir/tenofovir alafenamide/emtricitabine (BIC/TAF/FTC) on day +46, dermatological, gastrointestinal, and respiratory signs and symptoms persisted. It is of note that mpox PCR on mucocutaneous lesion exudate remained positive on day +51. Due to progressive hypoxemia, bronchofibroscopy with bronchoalveolar lavage (BAL) collection was repeated on day +67. Besides positive mpox PCR on BAL, all other microbiological exams in this product were negative.

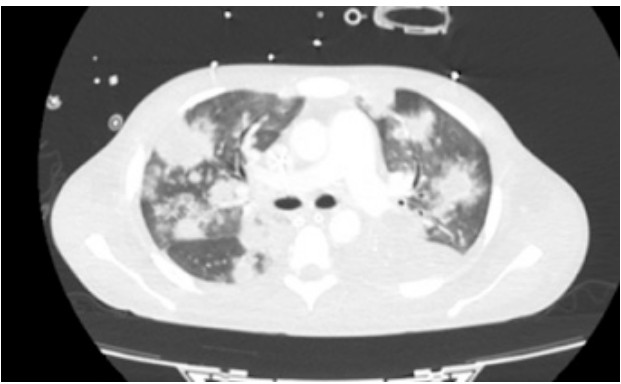

**Figure 2.** Thoracic CT scan on day +43 demonstrating multiple homogeneous nodular lesions scattered throughout the parenchyma.

On day +68, the patient was transferred to a tertiary hospital in Lisbon. At that moment, he presented polypnea, severe hypoxemia, and a high-volume left pleural effusion that was successfully drained. Mpox pleural fluid PCR was positive. Oral tecovirimat therapy was started the next day, and broad-spectrum antibiotics were maintained, but due to type I respiratory failure refractory to supplementation with fraction of inspired oxygen of 100%, orotracheal intubation was performed, and mechanical ventilation (MV) was started on day +71.

On ICU admission, cranial CT scan was unremarkable. In addition, LP on day +77 documented increased opening pressure (26 cm $H_2O$) and normal cytochemical features along with positive mpox PCR in CSF.

Despite being on ARV therapy since day +46 as previously mentioned, during an approximate three-week interval, HIV viral load increased from 36,904 (day +9) to 795,000 copies/mL (day +68). Additionally, on D6 of oral tecovirimat therapy, there was progression of skin ulcerations and extensive destruction of the perineum and gluteal region, with evidence of fistulization to rectum and prostate on abdominopelvic CT. For this reason, a protective colostomy was performed on day +73 to prevent contamination of the perineal lesions and allow for cicatrization to occur. In addition, CT scan showed jejunal and colic increased mucosal contrast uptake, suggesting enterocolitis (Figure 3). These facts raised concerns about probable insufficient enteric absorption of ART and tecovirimat considering

the progressive gastric content stasis and enterocolitis documented on CT and therefore possible extension of inflammatory process to small bowel. This situation prompted the search of a parenteral option to both medications. On day +73 intravenous (IV), cidofovir (5 mg/kg once weekly for 2 weeks, then once every other week, following WHO interim guidance June 2022) was added to oral tecovirimat (which completed 21 days) as antiviral therapy for mpox. HIV viral load increased to 5,000,000 copies/mL, and TCD4+ dropped to 38/μL on day +75, and there was confirmation of no other mutation than M184V in contemporary resistance testing along with no mutations detected for the integrase gene. Together with viral enteropathy—the tissue had abundant inflammatory cell infiltrate and villous atrophy—reported in duodenal biopsy on day +86, which was also PCR-positive for mpox, confirmed the likely lack of absorption of oral BIC/TAF/FTC. A full parenteral regimen with intramuscular (IM) cabotegravir (CAB) and IM rilpivirine (RPV) was started on day +95, both to be administered monthly, with initial dosing of 600 mg of CAB and 900 mg of RPV together with IV zidovudine 50 mg every four hours. Six days after the initiation of the later regimen, there was already a decrease in HIV viral load from 5,000,000 to 9010 copies/mL (Table 1).

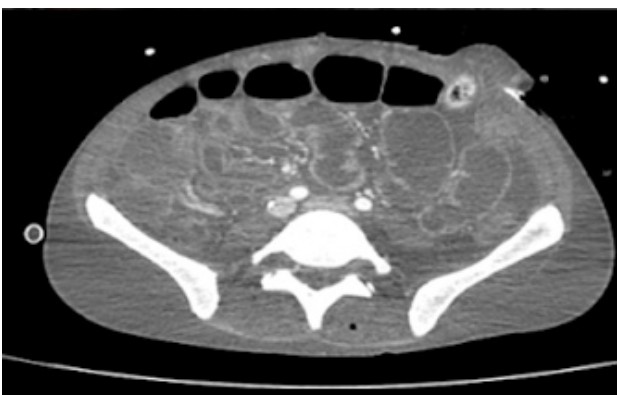

**Figure 3.** Abdominopelvic CT scan on day +73 with marked thickening of jejuno-ileum and colon mucosa, suggesting enterocolitis.

**Table 1.** Timeline of main events regarding ART modification of initiation and HIV restaging.

| Time Lapse | Events Regarding ART and HIV Infection Staging |
| --- | --- |
| 1 month prior to emergency department admission (Day 0) | Reported adherence to ART (dolutegravir/abacavir/lamivudine) after a 12-month therapy withdrawal |
| Ward admission (Day +9) | HIV viral load: 36,904 copies/mL; CD4+ count: 173/μL No ART |
| Day +46 | Bictegravir/tenofovir alafenamide/emtricitabine initiation |
| Admission in tertiary hospital (Day +68) | HIV viral load: 795,000 copies/mL |
| Day +75 | HIV viral load: 5,000,000 copies/mL; CD4+ count: 38/μL |
| Day +95 | IV cabotegravir/rilpivirine and IV zidovudine initiation |
| Day +101 | HIV viral load: 9010 copies/mL |

Despite therapeutic measures, clinical deterioration persisted. New mucocutaneous lesions developed, located on the borders of the first ulcers and isolated in the face, trunk, and limbs (Figure 4A–C). The ulcers documented at admission continued to progress in size and depth, being the most remarkable on the scalp (with epicranial aponeurosis exposure) and anal-gluteal-sacral region, with which, despite protective colostomy, there was further tissue destruction with rectal fistulization to the skin, which was evident on pelvic MRI on day +97 (Figure 5). With increasing doses of sedatives and analgesics, including epidural

catheterization and intrathecal morphine, pain management was still not successfully achieved, and MV weaning was not possible. Clinical frailty due to marked malnutrition, immunosuppression caused by AIDS, prolonged profound sedation, MV, and impairment of mucocutaneous barriers determined the occurrence of multiple infectious complications, namely Acinetobacter baumannii pneumonia and bacteremia. Despite appropriate therapy, the patient died of septic shock on day +107 upon admission and a concurrent HIV infection with acquired immunodeficiency syndrome (AIDS) criteria, in which severe small bowel involvement jeopardized normal therapy absorption for both conditions.

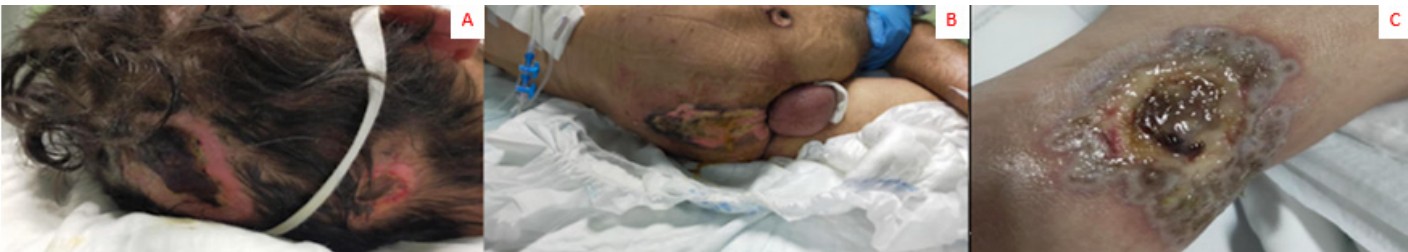

**Figure 4.** (**A–C**)New typical mucocutaneous lesions on ulcer borders, deep scalp ulcers, and gluteal-sacral deep ulcer (days 87–97).

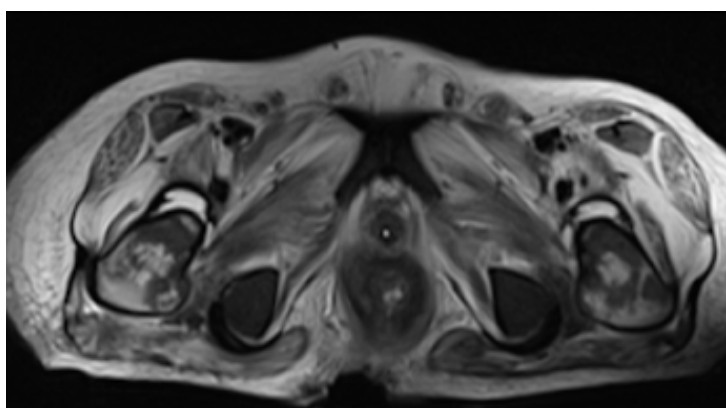

**Figure 5.** Pelvic MRI demonstrating deep gluteal-sacral soft tissues lesion causing severe barrier loss of integrity with rectal exposure.

## 3. Discussion

This patient developed an unusually progressive and disseminated disease with laboratory-confirmed involvement of CNS, lungs, gastrointestinal tract, and skin by mpox virus. This presentation is extremely rare in the context of the current outbreak, in which most patients experienced mild mucocutaneous disease with full resolution with no treatment in two to four weeks [9,10] This case is therefore rare in the mpox landscape and comparable only in course of disease duration (months) and progressive nature to the cases of oculocutaneous progressive gangrene reported by Carruba et al. [11] in two patients with AIDS and by Gowind et al. [3] in three patients also with AIDS.

This is the first documented case of disseminated mpox infection in Portugal with this severity and exuberance as well as the first national fatality by this cause. The most likely contributing factor for the course of the disease in this specific patient was the immunosuppressed status due to untreated and advanced HIV infection, as previously reported [12,13]. There are resemblances to the cases reported by Menezes et al. [14], Carruba et al. [10], and Gowind et al. [3] as for the link to advanced HIV infection. Nonetheless, the patient reported by those authors had not only a more depleted TCD4+ count (53/μL in Menezes et al.'s article; 55/μL and 29/μL in Carruba et al.'s article; 3/μL, <10/μL, and 71/μL versus 173/μL in our report), but Menezes' patient also had his last chemotherapy

cycle for lymphoma only two days before symptom initiation; therefore, a more severe state of immunodeficiency may have dictated clinical evolution to death in less than a month. As for our patient, the less depleted immune status may have been responsible for a slower yet constant clinical decline with addictive territory involvement and marked lesion progression in skin and gastrointestinal sites over more than five months of local and systemic therapeutic attempts. We interpret the appearance of new umbilicated lesions on the border of the first unhealed ulcers as a sign of ongoing viral activity on skin tissues, contemporary to cellular immunity progressive decline and refractoriness to the therapies instituted.

Carruba et al. [11] made a compelling point in their work comparing the progressive nature of the mpox disease of the two immunosuppressed patients reported to a formerly described orthopoxvirus disease manifestation known as progressive vaccinia or Vaccinia gangrenosum. This clinical entity was recognized in immunosuppressed patients following the administration of conventional smallpox vaccination. The disease is characterized by uncontrolled skin viral replication in the inoculation point, followed by viral dissemination to adjacent skin territories and profound tissues, where damaged tissue becomes necrotic, resulting in physiologic compromise of the tissue itself as well as bacterial superinfection and potentially sepsis and death. Indeed, this seems to be exactly the course of the disease in our patient, with monkeypox ulcers worsening in number, size, and depth over months and with a final fatal outcome as a result from sepsis caused by a nosocomial bacterial agent. It should also be added that, opposed to the observations made in Carruba et al.'s reported cases, in which there was no viral identification in primary sterile sites non-contiguous to skin or eye structures, our patient had viral identification in BAL, pleural fluid, duodenal biopsy, and CSF. This fact reinforces a physiopathological mechanism in which viraemia has a primary role explaining viral invasion of sterile sites, which is in contrast to the clinical findings in Carruba et al., and is better explained by a process of direct inoculation and local necrosis and progression. Even though the disease progression mechanism seems different, the fatal event—namely superinfection by bacterial pathogen and sepsis—was the same in our patient and at least one of the patients described by Carruba et al.

As immunity status seems fundamental to control mpox infection, the inability to achieve HIV suppression and immune reconstitution was therefore a critical insufficiency in the process of treatment of this patient. We should acknowledge that there was a late initiation of ART, even considering the typical two week deferral for most opportunistic infections. Considering that treatment for suspected CMV colitis was performed, ART should have been more promptly initiated in a moment that absorption (a central problem in this case, as discussed latter) would be more likely successful, as in an early course of disease, absorptive structures might have been more spared.

HIV viral load increase and TCD4+ lymphocyte count drop, together with consistent molecular demonstration of lack of HIV resistance to the oral drugs administered, suggested lack of absorption of antiretrovirals. This phenomenon is supported by radiological evidence of bowel inflammation and by the demonstration of viral enteropathy on duodenal biopsy (with mpox DNA amplification in this tissue). Anatomopathological findings in duodenal biopsy—inflammatory cell infiltrate and villous atrophy—although unspecific, are those displayed in other diseases, such as celiac disease and tropical spue, in which chronic inflammation leads to a malfunctioning (atrophied) intestinal epithelial lining with impaired capacity of absorption of nutrients and drugs [15]. Therefore, this may be the first documented case in the literature so far of mpox enteropathy, with reports of gastrointestinal involvement so far concerning only distal affection, such as proctocolitis [16,17]. We hypothesize mpox enteropathy as one of the most important factors associated with worse outcome in this case since it precluded normal first-line therapy absorption for both conditions.

Reasonable doubt of effective absorption prompted the search for a full parenteral ARV regimen and anti-mpox parenteral therapy as well. Currently, the only approved parenteral drug for HIV-1 treatment is ZDV, which is ineffective in monotherapy. This

prompted us to assort a regimen with long-acting (LA) IM CAB/RPV, with this combination being approved in some European countries for virologically suppressed patients and after an oral lead-in period—none of which were conditions met in this case. Due to urgency in HIV virological replication control and the inexistence of other viable options with a pharmacokinetic reassuring profile, a regimen with IM CAB/RPV and IV ZDV was approved by local health authorities. Although effectiveness evidence is scarce, long-acting IM CAB/RPV has been previously successfully used in patients with HIV-1 detectable viral load. In a 2022 study by Christopoulos [18], a regimen with LA IM CAB/RPV was initiated in 15 patients with detectable viremia (median CD4$^+$ cell count, 99 cells/mm$^3$; mean log10 viral load, 4.67; standard deviation, 1.16). As a result, 12 (80%; 95% CI, 55–93%) achieved viral suppression, and the other 3 had a 2-log viral load decline by a median of 22 days. Indeed, our patient had a remarkable (~2.5-log) HIV-1 viral load decline in only six days. Assuming that IV ZDV use could not explain by itself such a virological decrease, part of the effect on the viral load must be attributed to the use of IM CAB/RPV. Although ultimately futile prognosis wise, the magnitude of the success in viral load decrease with this regimen must be pointed out.

Management of mpox disease should involve early antiviral therapy with tecovirimat for selected patients, particularly if they are immunocompromised and/or with severe infection [19]. Although there are still no data on human effectiveness in treating mpox—a recent study in the USA with 596 patients showed fair tolerability but no comparator was established to draw efficacy conclusions [20]—Gowind et al. offers remarkable insight on what might be the low efficacy of tecovirimat in extremely immunosuppressed individuals, as the drug was used in all three patients with no response [3]. In the case we report, it is not obvious that non-response to tecovirimat was due to non-absorption or the primary drug's low effectiveness in a patient with such severe clinical presentation and immunosuppression; there is very little experience of tecovirimat use in critical mpox patients. Cidofovir use followed the same rationale concerning potential absorption issues. However, despite being one of the recommended options [9] for treatment of severe mpox, evidence of cidofovir efficacy in controlled trials stems only from animal studies.

## 4. Conclusions

This case is intended to provide novel information on unusually severe clinical manifestations of mpox, namely proven extensive mucocutaneous, pleuropulmonary, gastrointestinal, and CNS disease, with unusually long clinical progression over months in the context of immunosuppression. This course of disease has been rarely reported in the current mpox outbreak and has strong resemblances to progressive vaccinia, a disease occurring in immunosuppressed hosts and caused by uncontrolled skin replication of orthopoxviruses used in smallpox vaccination.

This report also aims to alert physicians to the importance of timely recognition and prompt treatment initiation in case of severe and disseminated disease in the immunosuppressed patient.

Further studies are needed to better understand the predictable clinical efficacy of treatments such as cidofovir and tecovirimat in patients with disseminated and critical disease.

**Author Contributions:** J.C. and F.V.-L. wrote the manuscript; I.M., A.J., L.V.-F., H.P., D.P., F.M. and N.G. critically reviewed the manuscript. All authors have read and agreed to the published version of the manuscript.

**Funding:** The writing of this manuscript received the support of Fundação para a Ciência e a Tecnologia, grant number UIDB/04295/2020 and UIDP/04295/2020.

**Institutional Review Board Statement:** Not applicable due to non-experimental nature of the report.

**Informed Consent Statement:** Written informed consent was obtained from the patient's family for the publication of both clinical information and images since the patient could not express consent due to his clinical condition.

**Data Availability Statement:** Data available on request due to restrictions e.g., privacy or ethical.

**Conflicts of Interest:** The authors declare no conflict of interest.

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
