# Peer review of "Fatal Case of Progressive Mpox in a Patient with AIDS—Viral Enteropathy and Malabsorption Demanding the Use of Full Parenteral ARV and Endovenous Cidofovir"

_2036-7449, doi:10.3390/idr15020018_

Round 1

Reviewer 1 Report

Caria et al. report an interesting, fatal case of Mpox in a severely immunocompromised patient with HIV infection, with positive Mpox PCR in multiple sites, such as BAL, pleural fluid, CSF and duodenum. The presentation is interesting for its rarity so far, but the manuscript should be reviewed, as some sections need restructuring for clarity and the discussion needs revisiting. My suggestions are as follows.

Title

I suggest not including “(formerly Monkeypox)” in the title and adding it to the text (line 41).

Abstract

- Line 27 - I suggest removing “criteria”.

- Progression over months is mentioned on lines 26-27 and repeated on line 29. I suggest not repeating this information.

- Line 36 – I suggest removing “being to our knowledge the first documented case of enteropathy with this etiology” from the abstract.

Text

- Introduction – Govind et al. 2023 (“Severe Mpox Infections in People with Uncontrolled Human Immunodeficiency Virus (HIV)”) should be added as a reference.

- Lines 63-64 – Remove “the patient 63 was hospitalized”, as this is a repetition.

- Line 64 – Remove “nadir”. I assume the mentioned CD4 count is the CD4 count at the time he was hospitalized.

- Lines 64-65 – Change “ART was started and opportunistic infections were treated accordingly” to “Opportunistic infections were treated accordingly and ART was started”.

- Lines 67-68 – The authors mention the ART regimen was switched, but did not mention the previous regimen. I suggest mentioning the initial ART regimen.

- Line 75 – Please also specify the lost weight in Kg (in addition to the percentage).

- Line 86 – What oral antibiotic was started?

- Lines 86-87 – I suggest adding a sentence stating that the patient was discharged home.

- Line 94 – Add “and” before “C-reactive protein”.

- Line 98 – What were the “several regimens of empirical broad-spectrum antibiotics”?

- Line 99 – It seems that ART was started very late (day 46), even considering excluding/confirming opportunistic infections. What was the reason for this?

- Lines 95-124 – These two paragraphs are confusing and should be rearranged temporally.

- Lines 102-103 – Why was ganciclovir continued until day 30 if CMV viral load quantification in peripheral blood was negative and etiological study of colitis by other microorganisms did not yield positive results?

- Lines 104-105 – Please specify what microorganisms were searched in the “etiological study of colitis”.

- I assume the patient had a colonoscopy – please specify the colonoscopy findings.

- Line 118 – Add “and” before “mycobacterial”.

- Line 122 – Substitute “done” with “repeated”.

- Lines 127-128 – “The severity of this symptom mentally impaired the individual to the extent of him attempting suicide on two occasions”. How did the patient attempt suicide if he was hospitalized?

- Lines 128-129 – Was lumbar puncture performed because of the suicide attempts or because of sustained fever? Please clarify this.

- Line 130 – “the patient was transferred to a tertiary hospital in Lisbon”. Please include in the previous text where (what type of hospital) he was hospitalized until then.

- Line 133 – What does “partial” respiratory failure mean? I suggest removing “partial”.

- Line 135 – “Carried out” and not “carried away”. But I suggest substituting this verb for a more formal one.

- Lines 135-136 – “Mpox pleural fluid PCR was positive.” Please transfer this to line 132, after mentioning that the effusion was drained.

- Lines 138-139 – Please state why a repeat LP was performed.

- Line 153 – Why was oral tecovirimat maintained? Was probenecid added? Was there any nephrotoxicity? Why was oral ART initially maintained and only Mpox antiviral switched to IV?

- Line 157 – The positivity of Mpox PCR in the duodenal biopsy should not be stated between parentheses as this is not superfluous information.

- Line 161 – “Zidovudine 50 mg every 6 hours”. Is this the 1 mg/kg dosage? Is there a reason why it was used every 6 hours and not every 4 hours, as is recommended?

- Line 190 – Add “by Mpox virus” after “skin”.

- Discussion, first paragraph: Govind et al. 2023 (“Severe Mpox Infections in People with Uncontrolled Human Immunodeficiency Virus (HIV)”) should be added as a reference and discussed.

- Line 198 – Remove “probably” (the authors already state “The most likely” in this sentence).

- Line 213 – Substitute “makes” for “make” and “his” for “their”.

- Discussion: I think Carruba et al. do not report positive Mpox PCR in profound sites, such as in this case. In progressive vaccinia, death is usually due to bacterial superinfection of the skin and subsequent septic shock; I think, in this case, the possibility of direct viral involvement of several sites (PCR positivity in BAL, pleural fluid, CSF, duodenal biopsy, etc.) should be discussed. The possibility of a cytokine storm can also be considered.

- Lines 225-236 – I think the late initiation of ART should be acknowledged and the importance of early initiation of ART should be discussed (within two weeks for most opportunistic infections).

- Conclusion: “The lack of response to antiviral and antiretroviral therapy suggested the hypothesis of gastrointestinal involvement and possible Mpox enteropathy jeopardizing drug absorption. Switch to IV antiviral and ARV therapy were initiated, and although there was no clinical or laboratory evidence of Mpox infection improvement, we report an unexpectedly favorable HIV-1 viral load decrease.” I suggest removing these sentences, as this was already stated previously.

Figures

Figure 1 is a combination of photographs from several different scopes that should be separated into different figures. For example, photographs of the skin lesions can be grouped, but the thorax CT, the abdominal CT and the pelvic MRI should be different figures (one for each).

Author Response

RESPONSE TO REVIEWER 1 COMMENTS

Previous note: Some of the remarks made by the reviewer were not introduced in the manuscript as in some points was not clear if the editor only wanted clarification of some aspect of if that aspect should imperatively be metioned in the manuscript text2

Title

POINT 1: “I suggest not including “(formerly Monkeypox)” in the title and adding it to the text (line 41).”

RESPONSE 1: Changes were made in manuscript as suggested by the reviewer

Abstract

POINT 2: “- Line 27 - I suggest removing “criteria”.”

RESPONSE 2: Changes were made in manuscript as suggested by the reviewer

POINT 3: “- Progression over months is mentioned on lines 26-27 and repeated on line 29. I suggest not repeating this information.

RESPONSE 3: Changes were made in manuscript as suggested by the reviewer

POINT 4: - Line 36 – I suggest removing “being to our knowledge the first documented case of enteropathy with this etiology” from the abstract.

RESPONSE 4: Changes were made in manuscript as suggested by the reviewer

Text

- Introduction –

POINT 5:  Govind et al. 2023 (“Severe Mpox Infections in People with Uncontrolled Human Immunodeficiency Virus (HIV)”) should be added as a reference.

RESPONSE 5: Unfortunately the authors could not access such article – not on open access.

POINT 6:  - Lines 63-64 – Remove “the patient 63 was hospitalized”, as this is a repetition.

RESPONSE 6: Changes were made in manuscript as suggested by the reviewer

POINT 7:  - Line 64 – Remove “nadir”. I assume the mentioned CD4 count is the CD4 count at the time he was hospitalized.

RESPONSE 7: Changes were made in manuscript as suggested by the reviewer

POINT 8:  - Lines 64-65 – Change “ART was started and opportunistic infections were treated accordingly” to “Opportunistic infections were treated accordingly and ART was started”.

RESPONSE 8: Changes were made in manuscript as suggested by the reviewer

POINT 9:  - Lines 67-68 – The authors mention the ART regimen was switched, but did not mention the previous regimen. I suggest mentioning the initial ART regimen.

RESPONSE 8: Changes were made in manuscript as suggested by the reviewer. Reference to initial therapy with darunavir/cobicistat/tenofovir alafenamide/emtricitabine was added on line 72.

POINT 10:  - Line 75 – Please also specify the lost weight in Kg (in addition to the percentage).

RESPONSE 10: Changes were made in manuscript as suggested by the reviewer. The sentence “the patient lost 10kg from an initial weight of 65kg” was added on line 82.

POINT 11:  - Line 86 – What oral antibiotic was started?

RESPONSE 11: The patient was started on oral amoxicillin/clavulanic acid (875/125mg), one pill every 12h. The authors didn´t find relevant the inclusion of this information in the manuscript.

POINT 12:  Lines 86-87 – I suggest adding a sentence stating that the patient was discharged home.

RESPONSE 12: Changes were made in manuscript as suggested by the reviewer

POINT 13:  - Line 94 – Add “and” before “C-reactive protein”.

RESPONSE 13: Changes were made in manuscript as suggested by the reviewer

POINT 14:  - Line 98 – What were the “several regimens of empirical broad-spectrum antibiotics”?

RESPONSE 14: From day 0-day 25 empirical therapy with piperacillin/tazobactam and linezolid was carried out. From day 26-day 41:  Due to clinical worsening with maintained fever, extensive purulent and hemorrhagic perianal and rectal ulcerations with suggestion of perianal abscess and colitis in imaging, and respiratory failure with lung abscesses in imaging, antimicrobial therapy with linezolid was kept and gram-negative coverage was escalated to endovenous meropenem.

The authors didn´t find relevant the inclusion of this information in the manuscript.

POINT 15:  - Line 99 – It seems that ART was started very late (day 46), even considering excluding/confirming opportunistic infections. What was the reason for this?

RESPONSE 15: The exclusion/confirmation of opportunistic infections was indeed the main reason for differing the initiation of ART. Also, considering the history of suboptimal adhesion to antiretrovirals the clinical team opted for waiting on resistance testing prior to therapy initiation, which in this institution setting takes usually a lot of time to have the results back.

POINT 16: - Lines 95-124 – These two paragraphs are confusing and should be rearranged temporally.

RESPONSE 16: Extensive text rearrangement was done in order to describe the main events of the hospitalization in the first institution in chronological order. The way the case was described in the first manuscript was meant to offer a organ system oriented perspective (dermatological, respiratory, gastrointestinal…) with chronological description within each of the organ system problems.

POINT 17:- Lines 102-103 – Why was ganciclovir continued until day 30 if CMV viral load quantification in peripheral blood was negative and etiological study of colitis by other microorganisms did not yield positive results?

RESPONSE 17: The following sentence was added to the text, as explanation for that clinical attitude: “The high clinical suspicion and the late access to the result of negative CMV viral load justified that this therapy was carried until day+30”. We should also remember that at that time the clinical team had no experience with cases of mpox colitis of such severity, so it was considered that there was a higher probability of this presentation being justified by a known agent of colitis in an AIDS patient – CMV, in this cases – that by mpox virus itself.

POINT 18:- - Lines 104-105 – Please specify what microorganisms were searched in the “etiological study of colitis”.

RESPONSE 18:  Once hospitalized the following exams were carried away: bacterial coprocultures (with valorization of Campylobacter spp., Salmonella spp., Shigella spp., Clostridium difficile growth); feces bacterial and fungi antigen testing (including Giardia spp., Clostridioides spp., Isospora spp., Mycrosporidium spp. and Mycrosporidium); feces viral antigen testing (Norovirus, Astrovirus, Norwalk ); feces parasitological exam (search for eggs, cysts and parasytes);  Strongyloides spp., Entamoeba spp. and VDRL serologies; CMV serum PCR; Rectal exudate PCR for C. trachomatis and N. gonorrhoeae. Anal canal biopsy was negative for mycobacteria in direct, cultural and NAAT testing (the anatomopathological report of anal canal biopsy is included in the next response). Autoimmune markers related to inflammatory bowel disease were negative.

POINT 19:- I assume the patient had a colonoscopy – please specify the colonoscopy findings.

RESPONSE 19: Colonoscopy was attempted on day +25 but interrupted for lack of correct preparation. Further colonoscopic exams were avoided due to the risk of perforation (this was the reason invoked by the Gastrenterology team of the first hospital). Anuscopy was performed on day +17 with the following findings: “Deep perianal coalescing ulcerations were observed, with a devitalized bottom and hardened border (Monkeypox lesions?), with adjacent darkened skin areas suggestive of islets of spared skin. These ulcerations extend circumferentially through the anal canal and inferior rectum in a extension of about 5 cm in depth, with greater exuberance and infiltration at 7-8 am, without evidence

of rectal bulges on palpation. No apparent drainable collection.” Anal border lesions were biopsied on day+30: “- Anatomopathological Diagnosis:: Exanthematic anal lesion compatible with monkeypox virus infection in the pustular phase. However, due to the morphological similarity of the cutaneous lesions with other viral agents such as smallpox, cowpox, varicella zoster and herpes simplex, clinical and laboratory correlation is mandatory. -Macroscopic Description:: Circular flap of anal canal measuring 6 mm in diameter, in which a brown macula with ill-defined borders with a 5 mm long axis, less than 1 mm from one of the margins, can be identified. -Microscopic Description:: Tissue with focus of necrosis of the entire thickness of the epithelial layer, observing at the edge of the lesion spongiosis of the epithelial layer, ballooning degeneration of  these cells, other multinucleates, keratinocytes and the outline of spherical intracytoplasmic inclusions of Guarnieri, along with nuclei " empty". Direct exam, NAAT and mycobacterial culture of this tissue were negative”

POINT 20:- Line 118 – Add “and” before “mycobacterial”.

RESPONSE: Changes were made in manuscript as suggested by the reviewer

POINT 21:- - Line 122 – Substitute “done” with “repeated”.

RESPONSE: Changes were made in manuscript as suggested by the reviewer

 POINT 22- Lines 127-128 – “The severity of this symptom mentally impaired the individual to the extent of him attempting suicide on two occasions”. How did the patient attempt suicide if he was hospitalized?

RESPONSE: The patient tried to die by hanging with a bed blanket in the ward bedroom and the the second time made a neck cut with a glass he shattered in the ward.

POINT 23:- Lines 128-129 – Was lumbar puncture performed because of the suicide attempts or because of sustained fever? Please clarify this.

RESPONSE: Lumbar puncture was performed for behavior alteration (expressed through suicide attempts), maintained fever with elevated inflammation markers in a patient with confirmed mpox that could also have CNS involvement (meningoencephalitis) with possible cytochemical alterations in CSF – unfortunately, this first sample collected on day +34 was not sent for the National Reference Lab for mpox PCR. Th intent was also to test for other opportunistic agents in AIDS context: CSF was negative for bacterial cultural exam, mycobacterial direct, cultural and NAAT exams, JC polyomavirus PCR, herpesvirus 1-8 PCR, Toxoplasma gondii PCR and Cryptococcus neoformans antigen testing.

POINT 24- Line 130 – “the patient was transferred to a tertiary hospital in Lisbon”. Please include in the previous text where (what type of hospital) he was hospitalized until then.

RESPONSE: The patient was previously hospitalized in a district hospital in Barreiro. That information is now in the manuscript – line 80.

POINT 25- Line 133 – What does “partial” respiratory failure mean? I suggest removing “partial”.

RESPONSE: By partial the authors meant type I respiratory failure. That correction is now in the manuscript

POINT 26- Line 135 – “Carried out” and not “carried away”. But I suggest substituting this verb for a more formal one.

RESPONSE: The sentence was altered to “(…)orotracheal intubation was performed and mechanical ventilation (MV) was started on day +71”.

POINT 27:- Lines 135-136 – “Mpox pleural fluid PCR was positive.” Please transfer this to line 132, after mentioning that the effusion was drained.

RESPONSE: Changes were made in manuscript as suggested by the reviewer

POINT28:- Lines 138-139 – Please state why a repeat LP was performed.

RESPONSE: LP was performed in order to send CSF sample to the National Reference Lab for mpox PCR, to reevaluate the cytochemical properties of CSF and retest for opportunistic agents.

POINT 29:- Line 153 – Why was oral tecovirimat maintained? Was probenecid added? Was there any nephrotoxicity? Why was oral ART initially maintained and only Mpox antiviral switched to IV?

RESPONSE: Tecovirimat was maintained since this patient displayed a very ominous evolution and the clinical team wanted to maximize the probability of at of the two antivirals work. We are aware that there was no evidence to date of cidofovir and tecovirimat combination therapy in mpox.

Probenecid was added according to cidofovir protocol. There was no nephrotoxicity. Oral ART was maintained until IV formula was available – we needed a special authorization since cabotegravir/rilpivirine isn´t yet licensed in our country.

POINT 30:- Line 157 – The positivity of Mpox PCR in the duodenal biopsy should not be stated between parentheses as this is not superfluous information.

RESPONSE: Changes were made in manuscript as suggested by the reviewer

POINT 31:- Line 161 – “Zidovudine 50 mg every 6 hours”. Is this the 1 mg/kg dosage? Is there a reason why it was used every 6 hours and not every 4 hours, as is recommended?

RESPONSE: This sentence was mistaken: indeed the regimen performed was IV zidovudine 50 mg every 4 hours [6 times a day]. Correction was done in the manuscript

POINT 32- Line 190 – Add “by Mpox virus” after “skin”.

RESPONSE: Changes were made in manuscript as suggested by the reviewer

POINT 33- Discussion, first paragraph: Govind et al. 2023 (“Severe Mpox Infections in People with Uncontrolled Human Immunodeficiency Virus (HIV)”) should be added as a reference and discussed.

RESPONSE: Unfortunately the authors could not access such article – not on open access.

POINT 34- Line 198 – Remove “probably” (the authors already state “The most likely” in this sentence).

RESPONSE: Changes were made in manuscript as suggested by the reviewer

POINT 35- Line 213 – Substitute “makes” for “make” and “his” for “their”.

RESPONSE: Changes were made in manuscript as suggested by the reviewer

POINT 36- Discussion: I think Carruba et al. do not report positive Mpox PCR in profound sites, such as in this case. In progressive vaccinia, death is usually due to bacterial superinfection of the skin and subsequent septic shock; I think, in this case, the possibility of direct viral involvement of several sites (PCR positivity in BAL, pleural fluid, CSF, duodenal biopsy, etc.) should be discussed. The possibility of a cytokine storm can also be considered.

RESPONSE: The following considerations were added to the text in accordance to the reviewer´s suggestion: “It should also be added that, opposed to the observations made in Carruba et al. reported cases, in which there wasn´t viral identification in primary sterile sites non-contiguous to skin or eye structures, our patient had viral identification in BAL, pleural fluid, duodenal biopsy and CSF. This fact reinforces a physiopathological mechanism in which viraemia has a primary role explaining viral invasion of sterile sites, therefore in contrast to the clinical findings in Carruba et al., better explained by a process of direct inoculation and local necrosis and progression. Even though the disease progression mechanism seems different, the fatal event – namely superinfection by bacterial pathogen and sepsis – was the same in our patient and least one of the patients described by Carruba et al.”.

The hypothesis of cytokine storm as explanation to the apparently septic/systemic inflammation event does not seem to fit in the case we describe since there is a temporally plausible isolation of Acinetobacter baumanii in hemocultures that explains the processes that drove the fatal event.

POINT37- Lines 225-236 – I think the late initiation of ART should be acknowledged and the importance of early initiation of ART should be discussed (within two weeks for most opportunistic infections).

RESPONSE The following considerations were added to the text in accordance to the reviewer´s suggestion: We should acknowledge that there was a late initiation of ART, even considering the typical two week deferral for most opportunistic infections. Considering that treatment for suspected CMV colitis was performed, ART should have been more promptly initiated in a moment that absorption (a central problem in this case discussed latter) would be more likely successful as in an early course of disease absorptive structures might have been more spared.”

POINT 38- Conclusion: “The lack of response to antiviral and antiretroviral therapy suggested the hypothesis of gastrointestinal involvement and possible Mpox enteropathy jeopardizing drug absorption. Switch to IV antiviral and ARV therapy were initiated, and although there was no clinical or laboratory evidence of Mpox infection improvement, we report an unexpectedly favorable HIV-1 viral load decrease.” I suggest removing these sentences, as this was already stated previously.

RESPONSE Changes were made in manuscript as suggested by the reviewer

Figures

POINT39 Figure 1 is a combination of photographs from several different scopes that should be separated into different figures. For example, photographs of the skin lesions can be grouped, but the thorax CT, the abdominal CT and the pelvic MRI should be different figures (one for each).

RESPONSE Changes were made in manuscript as suggested by the reviewer

Reviewer 2 Report

The authors report a fatal case of disseminated Mpox infection and HIV infection with AIDS criteria accompanied by enteropathy and intestinal drug malabsorption. The authors provide a detailed description of the clinical course of the case. Currently, several questions about the severity and mortality of Mpox in immunocompromised individuals remain to be elucidated, which is why the subject is of interest and of utmost importance.
My recommendations and suggestions are:

- In section "2. Case Report", I would recommend describing the case in chronological order, from day +0 of admission to day +107, as well as mentioning in detail the diagnoses on admission, transfer to a tertiary hospital, admission to ICU and death.

- I would recommend making a figure that graphs the evolution of the  CD4+ T-cell count, viral load (HIV-1) and mark when the changes in antiretroviral therapy occurred, and if possible try to relate it to the most important and relevant clinical data at those times.

- Regarding figure N°1, I would recommend to separate the photos of the lesions from the MRI and CT images, or in any case to join those photos of lesions with images of concontemporaneous radiological studies in a single section (A or B or C...) and to order it chronologically.

- It would be interesting to deepen more in the physiopathology of the possible enteropathy caused by the monkeypox virus.

Author Response

RESPONSE TO REVIEWER 2 COMMENTS

POINT1: - In section "2. Case Report", I would recommend describing the case in chronological order, from day +0 of admission to day +107, as well as mentioning in detail the diagnoses on admission, transfer to a tertiary hospital, admission to ICU and death.

RESPONSE: Extensive text rearrangement was done in order to describe the events in chronological order. Also more detail was added to the diagnoses in the mentioned moments.

POINT2: - I would recommend making a figure that graphs the evolution of the  CD4+ T-cell count, viral load (HIV-1) and mark when the changes in antiretroviral therapy occurred, and if possible try to relate it to the most important and relevant clinical data at those times.

RESPONSE: A table was made to illustrate the points cited.

POINT3:- Regarding figure N°1, I would recommend to separate the photos of the lesions from the MRI and CT images, or in any case to join those photos of lesions with images of concontemporaneous radiological studies in a single section (A or B or C...) and to order it chronologicaly.

RESPONSE: Changes were made in manuscript as suggested by the reviewer

POINT4- It would be interesting to deepen more in the physiopathology of the possible enteropathy caused by the monkeypox virus.

RESPONSE: The following remarks both in the case report (line 328: “Together with viral enteropathy – tissue had abundant inflammatory cell infiltrate and villous atrophy - reported in duodenal biopsy on day +86, also PCR-positive for mpox”(…))and discussion (line 455: “Anatomopathological findings in duodenal biopsy – inflammatory cell infiltrate and villous atrophy – although unspecific, are those displayed in other diseases, like coeliac disease and tropical spue, in which chronic inflammation leads to a malfunctioning (atrophied) intestinal epithelial linning with impared capacity of absorption of nutrients and drugs [14].” ) were added.

Reviewer 3 Report

Dear editor, 

Colleagues from Lisbon describe the first Portuguese fatal case of disseminated mpox infection which progressed over more than three months in an AIDS patient with low cd4 cell count and uncontrolled HIV viral load. 

The case reporte appear to be of scientific interest and raise concerns on the need to restore immune recovery in AIDS patients coinfected with mpox. The need of effective oral and intravenous antivirals for severe mpox cases is even clarly stated.

Only minor editing and English revision is needed. I present few of them as  not definitive examples:

1. abstract line 27 i will delete criteria 

2. Case report line 128 Lumbar puncture (LP) 

Author Response

Dear reviewer,

Thank you for your suggestions.

The English was thoroughly reviewed by a native speaker after your remarks

Round 2

Reviewer 1 Report

I maintain the following suggestions:

- Introduction – Govind et al. 2023 (“Severe Mpox Infections in People with Uncontrolled Human Immunodeficiency Virus (HIV)”) should be added as a reference.

- Discussion, first paragraph: Govind et al. 2023 (“Severe Mpox Infections in People with Uncontrolled Human Immunodeficiency Virus (HIV)”) should be added as a reference and discussed.

- Lines 104-105 – Please specify what microorganisms were searched in the “etiological study of colitis”. The authors answered this question in their response letter but this information should be included in the manuscript.

- I assume the patient had a colonoscopy – please specify the colonoscopy findings. The authors answered this question in their response letter but this information should be included in the manuscript.

- Line 161 – “Zidovudine 50 mg every 6 hours”. Is this the 1 mg/kg dosage? Is there a reason why it was used every 6 hours and not every 4 hours, as is recommended? RESPONSE: "This sentence was mistaken: indeed the regimen performed was IV zidovudine 50 mg every 4 hours [6 times a day]." This is still not corrected in the manuscript.

Author Response

The document attached includes the responses to the points raised by Reviewer 1

Reviewer 2 Report

The authors resolved the doubts and suggestions given above. I have no further comments.

Author Response

The authors thank Reviewer 2 for the points raised.